# Epidemiological Review of Porcine Reproductive and Respiratory Syndrome Virus (PRRSV) in Japan: From Discovery and Spread to Economic Losses and Future Prospects

**DOI:** 10.3390/vetsci12060554

**Published:** 2025-06-05

**Authors:** Osamu Taira, Atsushi Kato, Nobuyuki Tsutsumi, Katsuaki Sugiura

**Affiliations:** 1General Incorporated Foundation Nippon Institute for Biological Science, Tokyo 198-0024, Japan; 2Nisseiken Co., Ltd., Tokyo 198-0024, Japan; 3Graduate School of Agricultural and Life Sciences, The University of Tokyo, Tokyo 113-8657, Japan

**Keywords:** PRRSV (porcine reproductive and respiratory syndrome virus), epidemiology, Japan, molecular epidemiology, control strategy

## Abstract

Porcine reproductive and respiratory syndrome virus causes a disease that inflicts significant economic losses to Japanese swine industry. This review article summarizes the history of this virus in Japan since its first detection in 1993 to the current situation, aiming to clarify the challenges for control and inform improved future management strategies. Findings indicate that the virus has diversified over time, with more virulent strains and new genotypes from overseas emerging. Strains related to widely used vaccines have recently become prevalent domestically, and current vaccines do not always provide sufficient protection against all circulating variants, raising safety concerns. Continuous monitoring is crucial, as the virus evolves rapidly, complicating control efforts. In conclusion, reducing the disease’s impact requires an integrated approach combining improved vaccines, strict on-farm hygiene (biosecurity), ongoing surveillance of viral genotypes, and coordinated regional cooperation among farms. This research contributes to the development of effective measures to protect the Japanese pork supply and the economic viability of Japan’s pig farms.

## 1. Introduction

Porcine reproductive and respiratory syndrome (PRRS), caused by the PRRS virus (PRRSV), is one of the most economically significant viral diseases affecting the global swine industry [1,2]. The disease is characterized by reproductive failure in late-gestation sows, including abortions, premature farrowing, increased stillbirths and mummified fetuses, and the birth of weak piglets, as well as respiratory distress, poor growth, and immunosuppression in pigs of all ages, particularly during the nursery and fattening stages [2]. Owing to its substantial economic and social impact, PRRS is listed by the World Organisation for Animal Health (WOAH, formerly OIE) [3], consequently affecting international animal quarantine and the pork trade. The estimated annual economic losses are immense, reaching approximately USD 664 million in the United States [4] and an estimated JPY 28 billion (USD 300 million) in 2009 in Japan [5,6], thereby causing significant damage to swine production worldwide.

PRRSV, a positive-sense single-stranded RNA virus, belongs to the family *Arteriviridae* of the order *Nidovirales* [1]. Its genome is approximately 15 kb long and contains at least ten open reading frames (ORFs). The virus exhibits remarkable genetic diversity and is broadly classified into two major genotypes, Type 1 (European genotype, *Betaarterivirus suid 1*) and Type 2 (North American genotype, *Betaarterivirus suid 2*), which share only approximately 60% nucleotide identity between them [1]. Furthermore, each genotype possesses a high mutation rate (estimated at 10^−2^ to 10^−3^ substitutions/site/year [7]) and comprises numerous genetic lineages and clusters. This extensive genetic diversity and rapid evolution pose major obstacles to the development of effective control measures and broadly cross-protective vaccines [8,9]. Moreover, host genetic factors significantly influence the porcine response to PRRSV infection, adding another layer of complexity to disease control [10].

PRRSV primarily infects porcine alveolar macrophages (PAMs), monocytes, and dendritic cells [2] and employs sophisticated mechanisms to evade the host immune response. These mechanisms include the suppression of type I interferon (IFN-I) production, delayed induction of neutralizing antibodies (NAbs), potential antibody-dependent enhancement (ADE) of infection, and induction of immunosuppressive cytokines such as IL-10 [11]. Such immunosuppression, particularly the impairment of PAM function (PAMs are the primary target cells for PRRSV) renders pigs highly susceptible to secondary infections by other pathogens, notably bacteria such as *Streptococcus suis*, *Haemophilus parasuis*, and *Mycoplasma hyopneumoniae*. This often consequently leads to the exacerbation of clinical disease as part of the Porcine Respiratory Disease Complex (PRDC) [12]. Such evasion contributes to persistent infection, often lasting several months, and further heightens their vulnerability to these secondary infections, which frequently occur as part of the PRDC, wherein PRRSV interacts with other respiratory pathogens [12]. The sophisticated mechanisms by which PRRSV evades the host immune system have been comprehensively reviewed elsewhere [13]. Recent studies have revealed that specific viral proteins, such as Nsp5, interfere with key innate immune signaling pathways, including the cGAS-STING pathway [14].

In Japan, PRRS remains a primary challenge for the swine industry. Following suspected cases in the late 1980s [15] and initial clinical outbreaks around 1990 in the Kanto region [15], PRRSV was first isolated domestically in 1993 [16,17]. The initial isolates were predominantly Type 2 [16,18], which has since become endemic. Although genetically relatively limited strains circulated initially [18], viral genetic diversity has increased over time [19]. This increase includes the incursion of novel genetic lineages from abroad, such as Type 1 [20] and virulent Type 2 strains [21]. Currently, multiple genetic clusters co-circulate within Japan, with the dominant types varying by region and time (Figure 1) [22]. Furthermore, the circulation of vaccine-derived strains from widely used modified live virus (MLV) vaccines, coupled with the risk of recombination between vaccine and field strains, poses significant concerns [22,23].

This review aims to provide a comprehensive and chronological overview of the epidemiology of PRRS in Japan, from its initial discovery to the present, based on the available scientific literature. Specifically, it will cover (1) the discovery of PRRSV and the characteristics of early genotypes (clusters) in Japan; (2) the introduction of Type 1 virus and virulent strains, leading to increased genetic diversity; (3) the impact of PRRS infection on production performance and its estimated economic losses; (4) recent trends in viral circulation (changes in cluster composition) and their association with vaccine use; (5) the latest risk posed by the incursion of foreign strains (e.g., NADC34-like virus); (6) challenges related to vaccines and preventive measures; and (7) examples of regional control efforts in Japan and future prospects. By organizing and synthesizing this information, this review seeks to clarify the current status and challenges of PRRS control in the Japanese swine industry and contribute to the development of more effective future control strategies.

## 2. Discovery of PRRSV and Characteristics of Early Genotypes in Japan (1993–2007)

The history of PRRS in Japan dates back to the late 1980s. Antibodies against PRRSV were detected in sera collected from domestic pigs in 1987 [15], suggesting that the virus might have been present before clinical outbreaks were recognized. The first clinical reports emerged around 1990 from multiple farms in the Kanto region [15], describing collective reproductive failures (e.g., abortions, stillbirths) and respiratory problems in piglets. These clinical manifestations closely resembled the emerging disease syndromes reported in North America (“Mystery Swine Disease”) and Europe (“Blue-eared pig disease”) at that time [1,2]. Subsequent investigations led to the first official isolation of PRRSV in Japan in 1993 from pigs exhibiting “Heko-Heko disease” on a farm in Chiba Prefecture (strain EDRD-1) [16,17].

From the beginning, the PRRSV strains detected in Japan were predominantly of the North American type (Type 2) [16,18], potentially reflecting the origins of imported breeding stock during that period. Yoshii et al. [18] analyzed the *ORF5* gene (encoding the major envelope glycoprotein GP5, a highly variable region utilized for phylogenetic analysis) of 30 PRRSV isolates collected from 23 prefectures in Japan between 1992 and 2001, confirming that all were NA-type. Their phylogenetic analysis, including global NA-type strains, revealed that NA-type viruses could be broadly classified into five major genetic clusters (Cluster I–V). This analysis further showed that a majority (20 out of 32, 63%) of the Japanese isolates belonged to a specific cluster, designated Cluster III [18]. Although Cluster III included some isolates from China and Taiwan, it was considered a distinct genetic lineage predominantly established within Japan. Furthermore, Cluster III isolates exhibited a tendency for greater geographical distribution in the eastern regions of Japan [18]. However, isolates belonging to Cluster I, which includes US isolates and the PrimePac^®^ vaccine strain, were also present among the 1992–1993 Japanese isolates but were primarily found in the western regions of Japan [18]. By the 2000–2001 period, isolates belonging to Cluster II (containing the Ingelvac^®^ PRRS MLV vaccine strain and strains from the US and Asia) and Cluster V were detected, albeit in small numbers, along with some unclassified strains [18]. These findings indicated a gradual increase in the genetic diversity of PRRSV within Japan during the early 2000s [18,19].

Iseki et al. [19] compared ORF5 sequences from Japanese PRRSV isolates collected during three periods (1992–1993, 2000–2001, and 2007–2008) and confirmed a trend of increasing genetic heterogeneity over time. In particular, the nucleotide identity among Cluster III isolates decreased from an average of 95.7% (relative to reference strain EDRD-1) in 1992–1993 to 91.2% in 2007–2008, indicating ongoing genetic divergence within this cluster. While Cluster III remained the most prevalent cluster in 2007–2008 (accounting for 12 out of 17 isolates, 71%), Cluster II isolates were detected more frequently (3 out of 17, 17%) [19]. Many of these Cluster II isolates were genetically very close to the Ingelvac^®^ PRRS MLV vaccine strain, which was widely used in Japan at that time, suggesting the potential circulation and establishment of vaccine-derived strains in the field [19]. These findings highlight the establishment of the NA-type virus in Japan, the presence of a unique predominant domestic lineage (Cluster III), and increasing genetic diversity, including the emergence of Cluster II—potentially linked to vaccine use—by the late 2000s.

## 3. Incursion of European Type and Expansion of Genetic Diversity (2008–2013)

Until 2007, all PRRSV strains identified in Japan were Type 2. However, the genetic landscape of PRRSV in the country became significantly more complex in the subsequent years. Iseki et al., in their survey conducted during 2007–2008, identified a novel genetic lineage distinct from Clusters I, II, III, and V [19]. This new lineage, represented by the isolate Jpn5-37, was designated Cluster IV [19]. The *ORF5* gene of Jpn5-37 shared a high nucleotide identity (97.1%) with MN184A, a virulent strain isolated in Minnesota, USA, in 2001 [21]. Further analysis by Iseki et al. [21] revealed that the full genome of Jpn5-37 shared 93.6% identity with MN184A, exhibiting particularly high homology in ORF1b, ORF6, and ORF7 (91.8–97.3%). Crucially, the non-structural protein 2 (NSP2) gene region of Jpn5-37 exhibited three discontinuous deletions totaling 131 amino acids when compared to the VR-2332 prototype strain (111 aa + 1 aa + 19 aa deletions), a pattern identical to that found in MN184A [21]. The NSP2 gene is known to be the most variable region in the PRRSV genome, and specific deletions, particularly large ones such as the 111-aa deletions in MN184A/Jpn5-37, have been associated with increased virulence [21,24]. Yoshii et al. [25] had previously characterized the NSP2 of the endemic Japanese Cluster III strain EDRD-1, noting a unique 117-bp (39-aa) deletion and a 108-bp (36-aa) insertion (defined as nsp2-type E) relative to VR-2332. In contrast, Jpn5-37 and MN184A lacked this deletion/insertion pattern (belonging to nsp2-type V) but possessed distinct large deletions within NSP2 [21,25]. Animal experiments confirmed the higher pathogenicity of Jpn5-37 compared to EDRD-1, with Jpn5-37-infected animals exhibiting significantly higher viremia levels and more pronounced fever [21]. These genetic and pathogenic characteristics strongly suggested that Cluster IV represented a newly introduced virulent PRRSV lineage, likely originating from the US.

Furthermore, in 2008, the first isolation of European-type (Type 1) PRRSV in Japan was officially reported [20]. The isolate, Jpn EU 4-37, was found to be genetically related to Type 1 strains based on ORF5 sequence analysis. Animal experiments conducted by Iseki et al. [26] showed that while Jpn EU 4-37 alone caused only mild pathology [27], prior immunity to Type 2 PRRSV (EDRD-1) did not prevent infection with or viremia by Jpn EU 4-37. Conversely, prior Type 1 immunity did partially reduce viremia following Type 2 challenge [26]. These findings indicated that the introduction of the Type 1 virus posed a new epidemiological risk, even in herds endemic for the Type 2 virus.

Thus, from 2008 onwards, Japan faced a situation characterized by the co-circulation of diverse PRRSV strains: the endemic Japanese lineage (Cluster III/nsp2-type E) [18,25], vaccine-related lineages (Cluster II/nsp2-type V) [19], a virulent North American lineage (Cluster IV/nsp2-type V with deletions) [19,21], and the European type (Type 1) [20]. This rapid expansion of genetic diversity significantly complicated PRRS diagnosis, vaccine selection, and control strategies, heralding a new and more challenging phase for PRRS management in the Japanese swine industry.

## 4. Impact on Production Performance and Economic Losses (2006–2008 Survey and National Estimates)

The spread of PRRS has directly impacted the productivity of Japanese pig farms, posing a major economic burden. Several studies have quantitatively assessed this impact. Ishizeki et al. [28] conducted a cross-sectional study of 92 farrow-to-finish farms in 2010, analyzing the association between PRRSV status and production performance. Farm PRRSV status was determined based on routine serological (ELISA) and/or virological (PCR) testing under veterinary supervision. The study revealed a high prevalence of PRRSV-positive farms (76 out of 92, 82.6%). A comparison of production parameters revealed that PRRSV-positive farms had significantly higher post-weaning mortality (6.1% vs. 3.9% in PRRSV-positive and -negative farms; *p* < 0.01) and significantly lower average daily weight gain (585.1 g vs. 638.8 g; *p* < 0.01), resulting in considerably longer days to market (196.9 days vs. 180.3 days; *p* < 0.01). No significant differences were observed in carcass weight, feed conversion ratio, or reproductive parameters such as farrowing rate and litter size. These findings suggest that PRRS infection primarily affects the performance of growing-finishing pigs, potentially by reducing feed intake and growth rate, thereby necessitating extended feeding periods to reach market weight.

The decline in production performance inevitably translates into economic losses. Yamane et al. conducted a detailed study to estimate these losses. Based on data from six PRRS-affected farms [6], they calculated loss units per sow per day for specific clinical manifestations: JPY 22 for increased pre-weaning mortality, JPY 57 for increased post-weaning mortality, JPY 35 for increased finishing mortality, JPY 84 for abortions, JPY 30 for stillbirths, and JPY 32 for reduced growth rate. Abortions and post-weaning mortality were identified as having the largest economic impact [6]. Furthermore, Yamane et al. utilized these loss units, combined with data from a nationwide questionnaire survey of 121 farms (of which 65% reported experiencing PRRS outbreaks in the previous two years), to simulate annual economic losses. The simulation estimated a total annual loss of approximately JPY 3.525 billion for the 121 surveyed farms. Extrapolating this figure to the national sow population at that time (approximately 900,000 sows) resulted in an estimated nationwide annual economic loss of approximately JPY 28 billion (USD 300 million) [5]. This study was the first in Japan to quantify the substantial economic impact of PRRS, clearly demonstrating the severity of the damage [5,6,29]. The economic burden of PRRS encompasses not only direct losses from mortality and reduced growth but also increased costs for medication, vaccination, and disinfection, as well as losses due to secondary infections. Considering that this estimation was studied over 10 years before, it might not fully reflect the impact of recent circumstantial changes surrounding the pig industry and the emergence of new PRRSV strains that will be described later in Chapter 6. An updated reassessment is needed.

## 5. Shift in Cluster Composition and Association with Vaccines (2018–2020)

The genetic landscape of PRRSV in Japan continued to evolve into the late 2010s. Historically, PRRSV genotypes in Japan have been classified into Clusters I to V based on ORF5 sequences [18]. While this classification is widely utilized in domestic research and will be primarily employed in this review, the international standard is increasingly shifting towards a more detailed lineage classification system, such as that proposed by Yim-im et al. [30], which comprises 11 Lineages (L1–L11) and 21 Sub-lineages. The correspondence between these systems, according to Yonezawa et al. [31], is approximately as follows: Cluster I corresponds to Lineage 8, Cluster II to Lineage 5, Cluster III to Lineage 4, and Cluster IV to Lineage 1. Cluster V is also considered part of Lineage 4. Efforts are ongoing to refine PRRSV genetic classification systems globally to better track viral evolution and diversity, including proposals for dynamic nomenclature systems [30,32]. Bearing this correspondence in mind, this section discusses the recent shift in cluster composition in Japan and its potential association with vaccine use, focusing on the study by Kyutoku et al. [22].

Kyutoku et al. [22] analyzed ORF5 sequences from 2482 PRRSV-positive samples collected nationwide (32 prefectures, approximately 260 farms/year) between January 2018 and December 2020. It should be noted that these samples were collected under a diagnostic program initiated by a specific vaccine manufacturer and thus may not be fully representative of the entire Japanese pig population. Their extensive analysis revealed that Cluster II (accounting for 44.9–50.6% of total isolates) and Cluster IV (accounting for 34.0–40.8%) were the predominantly detected clusters during this period, collectively accounting for over 80% of the isolates [22]. In contrast, the proportion of Cluster III, previously the dominant endemic lineage in 2007–2008 [19], had significantly decreased to 7.8–12.1% (*p* < 0.01) [22]. Cluster I accounted for 3.1–6.7% of isolates and Cluster V for only 0.1–0.2% [22]. These finding indicates a major shift in the dominant PRRSV genotypes in Japan during the 2010s.

Significant regional variations in cluster distribution were also observed. In 2018, Cluster II was dominant across all regions. However, in 2019, Cluster II remained dominant only in the Hokkaido and Tohoku regions, while Cluster IV became dominant in other regions (Kanto/Tosan, Tokai, Chugoku/Shikoku, and Kyushu/Okinawa). In 2020, Cluster IV was dominant in Kanto/Tosan and Kyushu/Okinawa, whereas Cluster II predominated elsewhere, illustrating a complex and dynamic epidemiological situation [22].

The study strongly suggested that the widespread use of MLV played a significant role in this shift [22]. Many farms included in the study utilized MLV vaccines, primarily Ingelvac^®^ PRRS MLV or Fostera^®^ PRRS. A large majority of the detected Cluster II isolates exhibited extremely high ORF5 sequence identity (average > 98%) to the Ingelvac^®^ PRRS MLV strain (derived from US strain VR-2332, belonging to Lineage 5A) [22]. This finding strongly suggests circulation of the vaccine strain itself, or slightly mutated derivatives thereof, or detection from vaccinated animals. Similarly, some Cluster I isolates showed high identity (97–100%) to the Fostera^®^ PRRS vaccine strain (derived from US strain P129, Lineage 8C) [22]. Fostera^®^ PRRS was commercially introduced in Japan in 2018, potentially linking its use to the detection of these Cluster I strains. Conversely, no isolates showed high homology to the PrimePac^®^ vaccine strain (Cluster I/Lineage 7), which is not approved for use in Japan [22]. These findings suggest that the long-term, widespread use of Ingelvac^®^ PRRS MLV and the more recent introduction of Fostera^®^ PRRS may be exerting selection pressure (i.e., vaccine pressure) on the domestic PRRSV population, favoring the spread of specific clusters (Clusters II and I) that are highly homologous to the vaccine strains [22]. While the increase in Cluster IV (Lineage 1) might be attributed to its high infectivity [21], its association with virulent strains such as MN184A (L1F) [21] and the NADC-like strains (L1A, L1C) warrants particular attention [30].

## 6. Incursion of Foreign Strains and New Risks (2024: Emergence of NADC34-like Strain)

The genetic diversity of PRRSV in Japan is influenced not only by domestic evolution and vaccine pressure but also by the constant risk of the incursion of novel viral lineages from abroad. Examples include the first confirmation of Type 1 virus in 2008 [20] and the emergence of Cluster IV (MN184A-like/Lineage 1F) [19,21]. In 2024, this risk materialized anew with the detection of another significant foreign lineage.

Yonezawa et al. [31] reported the first detection in Japan of PRRSV-2 strains belonging to Lineage 1, Sub-lineage 1.5 (alternatively Lineage 1A), from serum samples obtained from 18 piglets (42–56 days old) on a farm in Okinawa Prefecture. Phylogenetic analysis of the *ORF5* gene placed these Okinawan strains in a cluster closely related to the US NADC34 strain and NADC34-like strains isolated in Peru [33] and China [34]. The PRRSV NADC34 strain was first identified in the US in 2014 [35], and related NADC34-like (Lineage 1A) strains were found in other pig farming countries, indicating the rapid global spread. Severe reproductive losses in sow herds was well-documented in the US [30]. In China, NADC34-like strains have been increasingly detected since their first report around 2017, spreading to at least 10 provinces by 2021 and becoming one of the predominant epidemic strains in some areas [34]. The detection rate of NADC34-like strains in China soared from 3% in 2017 to 28.6% in 2021 [36]. These strains exhibit a variety of pathogenicity in China, with some NADC34-like isolates causing 100% miscarriage in experimentally infected pregnant sows but other isolates not [36]. The genetic diversity of NADC34-like strains in China was partly explained by the complex recombination events between the PRRSV NADC30-like strain and other strains, including the live attenuated PRRSV vaccines [37]. Similarly, some NADC34-like strains emerging via recombination between NADC34 (major parent) and NADC30 (minor parent) have been reported as a major cause of PRRS outbreaks in South Korea since around 2017, with some Korean isolates demonstrating high virulence in piglets [37,38]. The high transmissibility, virulence, and tendency for recombination of many NADC34-like strains highlight the potential difficulties in controlling the disease in countries where they have only recently been detected, such as Japan.

Analysis by Yonezawa et al. [31] showed that the *ORF5* gene of the Okinawan Lineage 1A isolates shared 90.9–92.9% (average 91.9%) nucleotide identity with the US NADC34 reference strain. This similarity was significantly higher than that with other Lineage 1 reference strains such as NADC30 (L1C), MN184 (L1F), or Jpn5-37 (L1F). Restriction Fragment Length Polymorphism (RFLP) analysis using the *Mlu*I-*Hinc*II-*Sac*II enzyme combination revealed that 10 of the 18 Okinawan isolates displayed the “1-7-4” pattern characteristic of NADC34, while the remaining 8 exhibited the “1-4-4” pattern often observed in NADC30 strains. This RFLP pattern distribution is consistent with reports for L1A strains (where 59.4% show the 1-7-4 and 11.4% show the 1-4-4 patterns), further supporting their classification as NADC34-like, although RFLP patterns alone are not definitive for genotyping. The prediction of N-linked glycosylation sites in the GP5 protein sequence identified glycosylation at asparagine residue 57 (N57) in 12 of the 18 Okinawan isolates (67%). This N57 glycosylation, along with the conserved N44 and N51 sites, is a characteristic feature that has been re-emerging in US L1A strains since 2015 [24,31]. The presence of strains lacking N57 glycosylation (six isolates) on the same farm suggests the co-circulation or evolution of multiple variants. Based on the combined evidence from phylogenetic analysis, RFLP patterns, and glycosylation site characteristics, Yonezawa et al. concluded that the isolates detected in Okinawa were NADC34-like strains (L1A), marking the first official report of this lineage in Japan.

NADC34-like strains pose a significant threat owing to their potential high pathogenicity and antigenic divergence from existing vaccine strains, raising concerns about amount of cross-protection afforded by the current vaccines [24]. Indeed, the development of new vaccines targeting NADC34-like strains is underway in countries such as China. Interestingly, the farm in Okinawa where these strains were detected reportedly experienced only a transient increase in piglet mortality and mild respiratory signs, without severe clinical manifestations like abortion storms that are typically associated with NADC34. This discrepancy could be attributed to the lower pathogenicity of the specific isolates, farm-specific environmental factors, co-infections, or the herd’s immune status, all of which require further investigation. Regardless of the observed pathogenicity in this specific case, the confirmed incursion of the NADC34-like lineage into Japan represents a serious warning for the nation’s PRRS control efforts. The potential spread of this virus to other regions could lead to significant damage, possibly overwhelming existing control measures. This finding underscores the urgent need to strengthen national PRRSV surveillance, particularly by monitoring for incursions by foreign strains, and to develop and update contingency plans for responding to outbreaks caused by novel, high-consequence lineages.

## 7. Challenges in Vaccines and Preventive Measures

Vaccination is a cornerstone of PRRS control worldwide. In Japan, while the primary vaccines currently approved and utilized are modified live virus (MLV) vaccines based on North American-type (Type 2) strains, such as Ingelvac^®^ PRRS MLV (Lineage 5A) and Fostera^®^ PRRS (Lineage 8C), inactivated vaccines (KVs) (Nisseiken PRRS Vaccine ME) are also available as an option (Table 1) [22]. While various types of PRRSV vaccines, mainly MLV vaccines and KVs, as listed in Table 1, are licensed in Japan, their adoption rates vary due to the year of licensing and commercial reasons. A 2017 survey by the Japan SPF Swine Association (71 responding farms) indicated that the implementation rate for PRRSV countermeasures (including vaccination) was approximately 30% on average, which was lower than for some other major swine diseases and showed regional variations [39]. However, it should be noted that this investigation was carried out in 2017 for a specific group of certified pig farms, so the current overall vaccine coverage across Japan remains to be fully elucidated.

MLV vaccines have demonstrated efficacy in reducing clinical signs (e.g., reproductive failure, respiratory disease), decreasing viral shedding, and shortening the duration of infection in numerous studies [8,9]. However, current MLV vaccines face several significant challenges and limitations: 1. Incomplete Protection: MLV vaccines do not provide sterilizing immunity; consequently, they cannot completely prevent infection, viral replication, or transmission. Vaccinated pigs can still become infected with field strains, exhibit mild symptoms, and shed virus, thereby acting as potential sources of infection [8,9]. 2. Safety Concerns: MLV vaccine viruses can persist in vaccinated animals for weeks to months, posing a risk of shedding and transmission to unvaccinated animals. Transmission via semen from vaccinated boars is a notable concern. Furthermore, vaccination of pregnant sows, especially in late gestation, carries a risk of the transplacental fetal infection of fetuses, potentially causing reproductive losses [8,9]. 3. Risk of Reversion and Recombination: As live viruses, MLVs can mutate during replication, presenting a theoretical risk of reverting to a more virulent form. A more significant concern is the potential for genetic recombination between MLV vaccine strains and circulating field strains (or even different MLV strains) when co-infection occurs [8,23]. Such recombination events have been reported globally [23] and can lead to the emergence of novel viruses with altered antigenicity or increased virulence, sometimes resulting in severe outbreaks. 4. Limited Heterologous Protection: Owing to the vast genetic diversity of PRRSV [18,19], immunity induced by one specific vaccine strain often fails to provide effective protection against genetically distant field strains [8,9]. This lack of broad cross-protection is a major limitation. For example, protection between Type 1 and Type 2 viruses is generally poor [26]. Even within Type 2, protection against different lineages (e.g., L1 vs. L5 vs. L8) can be insufficient [22,30]. This is particularly relevant with the emergence of novel strains such as NADC34-like strains (L1A), where existing vaccine efficacy may be compromised. The limited application of heterologous lineage virus protection by the current MLV vaccines is a particularly important issue in countries like Japan, where genetically divergent Lineage 1 PRRSV strains, which include NADC30-like, NADC34-like, and Cluster IV (Lineage 1F) strains, are prevalent. Studies evaluating the efficacy of VR-2332-based MLVs (the parent strain of Ingelvac^®^ PRRS MLV) against NADC30-like strains in China reported poor protection: vaccinated pigs showed similar lung lesions and viral loads in tissues to unvaccinated challenged pigs [40] or the vaccine showed limited efficacy against specific recombinant NADC30-like strains [41]. Similarly, a commercially available PRRSV vaccine showed incomplete protection against a highly pathogenic Chinese NADC34-like strain, with vaccinated pigs still showing viremia, pathological lesions, and mortality [42,43]. These findings highlight the efficacy issues of the current MLV vaccines against diversified PRRSV, particularly against Lineage 1 strains. In Japan, there remains a lack of published, direct comparative evidence evaluating the efficacy of the currently used MLV vaccines (e.g., Ingelvac^®^ PRRS MLV, Fostera^®^ PRRS) against prevalent domestic Cluster IV (Lineage 1F) strains or the recently detected NADC34-like (Lineage 1A) strains. Filling this knowledge with local data on key indicators like virus cross-neutralizing antibody titers and viral load suppression in the pig house is crucial for optimizing vaccine strategies in Japan and represents an important area for future research. Interestingly, reduced dosage strategies for some MLV vaccines are being explored for potential benefits against NADC34-like strains, although further investigation is required [44]. Experimental studies in Japan comparing Japanese Type 1 (Jpn EU 4-37) and Type 2 (EDRD-1) isolates revealed asymmetric cross-protection: prior Type 1 infection partially reduced Type 2 viremia, whereas prior Type 2 infection offered no protection against Type 1 challenge [26]. Conversely, prior infection with the endemic Japanese strain EDRD-1 (Lineage 4) did significantly reduce viremia and clinical signs upon challenge with a heterologous highly pathogenic PRRSV strain from Vietnam (Lineage 8) [45], suggesting some cross-protection is possible depending on the specific strains involved. 5. Delayed and Skewed Immune Response: PRRSV’s inherent immune evasion mechanisms can lead to the delayed induction of neutralizing antibodies and cell-mediated immunity, even following vaccination [8,11]. The induced response may also be skewed towards less effective Th2 immunity [11].

To address these challenges, global research efforts are focused on developing next-generation vaccines [8,9,46]. Inactivated vaccines, while generally safer, require improved immunogenicity, which is often pursued through advanced adjuvants (e.g., nanoparticles) and antigen processing techniques. Vector vaccines, which use other viruses or bacteria to deliver PRRSV antigens, offer the potential for enhanced safety and immunogenicity. Subunit vaccines, utilizing specific viral proteins or epitopes, provide high safety and design flexibility but necessitate potent adjuvants and delivery systems (e.g., baculovirus expression system, plant-based expression, nanoparticles, multi-epitope constructs). Nucleic acid vaccines (DNA and mRNA) represent a rapidly evolving platform, offering the potential for rapid development and the induction of both humoral and cellular immunity, although delivery and stability remain key areas for improvement [46]. Promising results have been observed with novel platforms like ferritin nanoparticle-based vaccines, which have demonstrated the ability to induce protective immune responses [47].

In the Japanese context, MLVs remain the primary tool, but their application requires careful consideration of each farm’s specific situation. Kawabata [48] demonstrated that, on a PRRS-endemic farm in a high-density area in Kagoshima, a two-dose MLV vaccination strategy (0.5 dose at 1 and 3 weeks of age) was more effective in reducing mortality and improving growth compared to a single dose administrated at 3 weeks. This finding suggested that early infection (pre-weaning) was occurring and that the two-dose regimen likely provided earlier immunity while potentially mitigating interference from maternal antibodies. This underscores the importance of understanding the infection dynamics on each farm (e.g., prevalent strain type, timing of infection) for optimizing vaccine selection and administration timing. Combination vaccination strategies, such as an MLV prime followed by an inactivated or DNA boost, are also being explored internationally to enhance immunity [46].

However, reliance solely on vaccination is insufficient for effective PRRS control. Stringent biosecurity is crucial for both preventing virus introduction and controlling its spread within the farm. Comprehensive, science-based biosecurity programs (often termed “Next Generation Biosecurity”) have demonstrated significant reductions in PRRSV incidence risk in large production systems [49]. Specific interventions such as air filtration and the use of feed mitigants have also shown protective effects against outbreaks [50]. These measures include external biosecurity (e.g., quarantine and testing of incoming animals, control of vehicle/personnel entry, disinfection, pest control) and internal measures (all-in/all-out pig flow, thorough cleaning and disinfection between batches, proper needle usage, movement control from contaminated to clean areas).

Accurate diagnosis and monitoring are also vital, yet the genetic diversity of PRRSV poses challenges to these efforts. Fukunaga et al. [51] highlighted that some conventional PCR primers designed before 2010 may fail to detect recent Japanese PRRSV strains owing to mutations in the primer binding sites. This emphasizes the need for updated diagnostic tools, such as their newly designed primer pair (PRRSV-M-2F/4R), to ensure reliable detection.

## 8. Future Actions and Prospects

Experience in Japan has demonstrated that controlling PRRS solely through individual farm efforts is often insufficient, particularly in densely populated pig production areas where the risk of virus re-introduction from neighboring infected farms via aerosols, personnel, vehicles, or other fomites is high [48,52]. Controlling PRRS is compounded by the geographical characteristics and the industrial structure of the swine industry in Japan, where pig farming houses are often concentrated in specific regions, leading to densely populated areas. High-density rearing environments increase the risk of viral transmission between farms and provide more opportunities for the virus to evolve and accumulate mutations through continuous circulation. Furthermore, in situations where MLV vaccines are widely used, vaccine strains and field strains could potentially interact to create and spread vaccine-derived recombinant viruses, thereby leading to the further genetic diversification of PRRSV in the country. Therefore, coordinated regional biosecurity measures using the molecular epidemiological surveillance of circulating strains has become even more critical. Consequently, regional control and elimination programs, defined as deliberate efforts to reduce disease incidence to locally acceptable levels or to achieve eradication within a defined area, are increasingly recognized as crucial strategies [53]. This understanding has led to a growing emphasis on regional control programs (RCPs) that involve coordinated efforts among multiple stakeholders within a defined geographical area.

A pioneering regional control initiative in Tahara City, Aichi Prefecture, Japan, starting with 98% of farms being positive for PRRSV in 2011 [52], demonstrated the importance of sustained and coordinated efforts. This multi-stakeholder program utilized the P-JET “PRRS Stage Classification System” (Figure 2) [54] for objective farm status monitoring and targeted interventions. Long-term follow-up reported significant progress: by the first half of FY2021, Stage 5 (negative/clean) farms increased from 0 (in FY2012) to 6 (14.3% of 42 tested farms) and Stage 1 (positive-unstable) farms were eliminated [55]. Concurrently, a shift in regional PRRSV genotypes was observed between FY2020 and the first half of FY2021, with a decrease in wild-type Cluster III strains (by 18%) and an increase in vaccine-related Cluster II strains (by 15%), suggesting a replacement of PRRSV with less pathogenic viruses owing to the extensive vaccination. Furthermore, individual farm case studies within the program indicated viral transitions from the wild-type strain to a vaccine strain following enhanced vaccination strategies, accompanied by improved production parameters such as increased number of born piglets and reduced post-weaning mortality. These improvements in infection status, viral genotype distribution, and production metrics underscore the benefits of such regional programs. Although detailed economic impact analyses for the entire region were not specified as the final outcome, the production gains clearly implied economic advantages. Sustaining motivation for the improvement of veterinary public health and addressing microbiological knowledge gaps remain key challenges for such initiatives. International experiences, such as those in the US, highlight the evolution of voluntary regional control programs and provide valuable insights into structuring effective collaborations and addressing associated challenges [56].

Further, a case study from Kagoshima Prefecture, one of the high-density PRRSV areas in Japan, reported that optimizing an MLV vaccination strategy within a specific farrow-to-feeder operation (two 0.5 doses at 1 and 3 weeks of age) markedly reduced the monthly piglet mortality rate from a maximum of approximately 30%/month to 4.3%/month, while also improving shipment weight from 37.1 kg to 38.9 kg and average daily weight gain from 417 g/day to 439 g/day [48]. This study suggested positive economic benefits in pig production gains. An overall decrease in post-weaning mortality across the managing organization’s farms was found (from 10.6% in FY2007 to 4.1% in FY2012). At the same time, this success indicated the limitations of individual farm-based efforts in a region where some farms are in densely populated areas. The detection of identical RFLP patterns of PRRSV among neighboring farms also highlighted the risk of farm-to-farm transmission in such areas, complicating control and prompting considerations for broader regional measures.

These Japanese examples underscore the limitations of farm-level control and the potential of coordinated regional strategies. P-JET [54] serves as a national platform facilitating such efforts by promoting science-based guidelines (including the Stage Classification) and supporting regional projects. Moving forward, advancing PRRS control in Japan will require scaling up these regional models and strengthening strategies in several key areas:Advanced Molecular Surveillance: Enhance nationwide surveillance by using not only ORF5 sequencing but also whole-genome sequencing (WGS) to rapidly detect genetic shifts, recombination events [23], virulence markers (e.g., NSP2 deletions) [21,25], antigenic changes, and transmission pathways [57]. The systematic analysis of large-scale sequencing data, as demonstrated in the US, is vital for tracking genetic evolution and identifying emerging strains [57,58]; dynamic classification systems can aid in monitoring these changes effectively. Continuous monitoring for emerging domestic variants and incursions of foreign strains, such as the NADC34-like virus, is critical.Strengthened Foreign Strain Risk Management: Maintain up-to-date knowledge of global PRRSV epidemiology, especially concerning Lineage 1 variants and HP-PRRSV. Bolster border biosecurity and quarantine measures. Improve early detection surveillance for foreign strains, while also refining contingency plans for rapid containment and eradication should incursions occur.To address these challenges, global research efforts are intensely focused on developing next-generation vaccines with improved safety and broader cross-protective efficacy [59]. Among nucleic acid vaccines, mRNA vaccine technology, which allows for rapid development, flexible antigen design, and induction of both humoral and cellular immunity, is emerging as a particularly promising platform for PRRSV. Recent studies have explored mRNA vaccines encoding various PRRSV antigens, such as GP5 alone or combinations like GP2-GP5-M polyprotein, which is easily encapsulated in lipid nanoparticles (LNPs) for efficient delivery and has demonstrated induction of significant antibody responses and T-cell activation in mice [59]. Self-amplifying RNA (saRNA) vaccines, which can achieve equivalent protection at lower doses than conventional mRNA, are also being investigated, though challenges in their delivery remain to be solved [59]. Subunit vaccines, utilizing specific viral proteins (e.g., GP3, GP5, M), epitopes, or even in silico designed immunogens, continue to be refined with advanced adjuvants and delivery systems (e.g., nanoparticles, virus-like particles, plant-based expression systems) to enhance their immunogenicity and protective breadth [59]. For example, ferritin-based nanoparticles displaying modified GP5 have been demonstrated to enhance Th1-type B cell immunity and protective effects in pigs [47,56].

Furthermore, strategies aimed at eliciting broadly neutralizing antibodies (bnAbs) by identifying and targeting conserved neutralizing epitopes across diverse PRRSV strains are a key focus. This approach includes the development of chimeric viruses, such as the rNADC34-CHSps candidate targeting NADC34-like PRRSV by Ye et al. (2025) [60], and reverse genetics to rationally design attenuated viruses with enhanced safety and immunogenicity profiles against recent field strains. Proteolysis-targeting chimera (PROTAC) technology has the potential for developing new generations of safer and broadly cross-protective live attenuated vaccines [59]. The ultimate goal is to develop vaccines that can overcome PRRSV’s extensive antigenic variability. This will provide robust and durable protection against the diverse array of prevalent viruses, including newly emerging highly pathogenic variants. The emergence of highly pathogenic NADC34-like PRRSV, now also detected in Japan, underscores the urgent need for efficient vaccines against these divergent strains, as the current commercial vaccines often provide incomplete protection and become a source of new recombinant viruses [61].

Since no NADC34-like specific vaccines are currently licensed anywhere, including Japan, international research efforts are underway. For example, Ye et al. recently reported the development of a live-attenuated chimeric vaccine candidate (rNADC34-CHSps) specifically targeting NADC34-like PRRSV [60]. This candidate was constructed by replacing the structural protein region of a NADC34-like strain (JS2021NADC34) with that of a CH-1R-like vaccine strain (CHR6) to improve its adaptation to Marc-145 cells, a common challenge in NADC34-like vaccine development [60]. This engineered chimeric virus showed stable replication in cell culture and, importantly, vaccinated piglets with this candidate induced significant protection immunity against a virulent NADC34-like strain, which led to reduced clinical symptoms, less severe microscopic pathological lesions, and lower viremia compared to unvaccinated controls. The development and evaluation of novel vaccines are needed in the face of continuously evolving field viruses.
4.Promotion and Support of Regional Control Programs (RCPs): Expand successful RCP models, such as Tahara’s [52], nationwide, adapting them to the local conditions. Robust RCP establishment and sustainability recommend a dual support approach, i.e., strengthening governmental financial and technical aid (e.g., for testing, culling, initial setup, emergency response), potentially via frameworks like the “Regional Chronic Disease Eradication Support Measures”, complemented by producer-led funding initiatives such as check-off systems. This dual approach is expected to bolster industry ownership and RCP resilience. Load-Close-Homogenize (LCH) strategies combined with vaccination also warrant consideration.5.Data-Driven Disease Control: Develop and implement integrated data management systems that combine genomic, farm management, host genetic [10], and production data. This will enable real-time risk assessment, outbreak prediction, and evaluation of control measure effectiveness.6.Continued Stakeholder Collaboration: Foster and maintain strong collaborative networks involving producers, veterinarians, government agencies, researchers, and industry partners (e.g., through P-JET). Such networks are essential for continuous information sharing, technology transfer, capacity building, and maintaining high biosecurity awareness.

## 9. Conclusions

This review has outlined the epidemiological trajectory of Porcine Reproductive and Respiratory Syndrome (PRRS) in Japan over approximately three decades, since the first domestic isolation of PRRSV in 1993 [16,17]. It has traced the dynamic evolution of the virus from the establishment of NA-type virus [16,18] to a unique endemic lineage (Cluster III) [18,25]. This evolution continued through a subsequent increase in genetic diversity (manifested by the rise of Clusters II and IV and the decline of Cluster III) [19,22], the incursion of the European type (Type 1) [20] and virulent NA-type lineages (Cluster IV/L1F) [19,21], and culminated in the most recent emergence of a significant foreign lineage (NADC34-like/L1A) [31]. Furthermore, this review examined the tangible impact of PRRS on production performance [28], its substantial economic burden (estimated at JPY 28 billion annually) [5], the challenges associated with widely used MLV vaccines (e.g., limited efficacy, safety concerns) [8,9,23,26], the complexities introduced by host genetic factors [10] and co-infections [12], and the growing importance of regional control initiatives [49,53,56] coupled with robust biosecurity measures [49,50].

PRRSV presents a formidable challenge owing to its high mutation rate [7], propensity for recombination [23], and sophisticated immune evasion strategies [11], rendering its control exceptionally difficult. The genetic pool of PRRSV in Japan has become increasingly complex, shaped by domestic evolution [18,19], vaccine-induced selection pressure [22], and incursions from abroad [20,21,32]. However, mitigating the impact of PRRS and progressing towards improved control is achievable through a multifaceted approach. This approach should combine advanced molecular epidemiology (e.g., phylogenetics, WGS) for understanding circulating strains, objective diagnostic methods [51], appropriate farm-specific strategies that integrate vaccination [48] (considering limitations against emerging strains [41,44]) and stringent biosecurity [49,50], and, crucially, coordinated regional efforts [49,52,55]. The recognition that farm-level interventions alone are insufficient, particularly in high-density areas, has spurred initiatives such as those in Aichi and Kagoshima prefectures. These initiatives demonstrate the potential of collaborative, science-based regional programs (e.g., utilizing Stage Classification [54]). Although eradication remains a long-term aspiration, effective PRRS control is an urgent national priority for the sustainability of Japan’s swine industry. Continued progress necessitates leveraging the latest scientific knowledge, as outlined in this review, alongside practical field applications driven by strong partnerships (including platforms such as P-JET). Strengthening surveillance, developing next-generation vaccines, enhancing biosecurity practices, exploring host genetic resistance, and fostering robust collaboration among all stakeholders are essential steps to effectively combat this persistent and costly disease.

## Figures and Tables

**Figure 1 vetsci-12-00554-f001:**
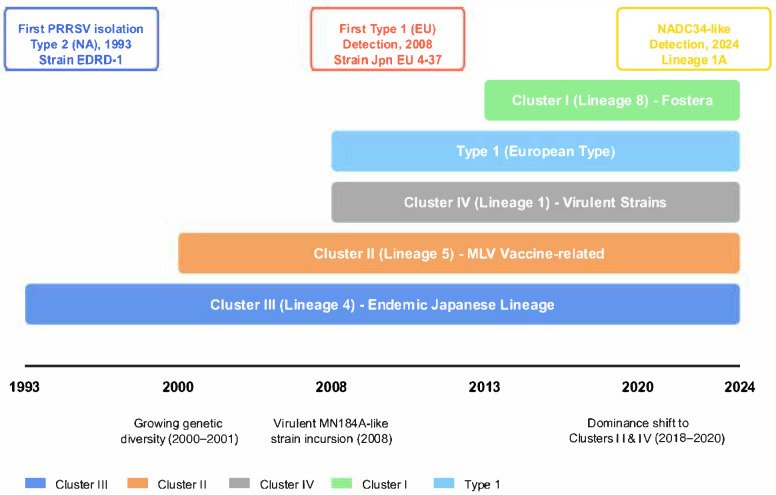
Genetic evolution of PRRSV in Japan (1993–2024). Timeline illustrating the genetic evolution of PRRSV in Japan since 1993, highlighting the emergence and shifts of major lineages and key incursions of novel strains.

**Figure 2 vetsci-12-00554-f002:**
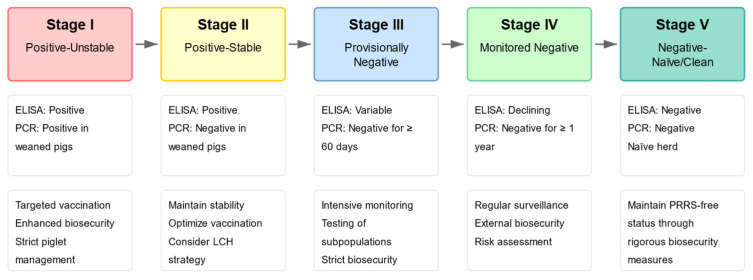
PRRS stage classification system for regional control programs. A five-stage PRRS herd classification system based on diagnostic status is used to guide interventions in Japanese regional control programs.

**Table 1 vetsci-12-00554-t001:** PRRS vaccines currently approved and used in Japan.

Vaccine Name	Type	Lineage	Manufacturer
Ingelvac^®^ PRRS MLV	MLV (Type 2)	Lineage 5	Boehringer Ingelheim (Ingelheim am Rhein, Germany)
Fostera^®^ PRRS	MLV (Type 2)	Lineage 8	Zoetis (Parsippany, NJ, USA)
Nisseiken PRRS Vaccine ME	KV (Type 2)	Lineage 4	Nisseiken Co., Ltd. (Tokyo, Japan)

## Data Availability

No new data created.

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
