# Peer review of "Epidemiological Review of Porcine Reproductive and Respiratory Syndrome Virus (PRRSV) in Japan: From Discovery and Spread to Economic Losses and Future Prospects"

_vetsci, 2025, doi:10.3390/vetsci12060554_

Round 1
Reviewer 1 Report
Comments and Suggestions for Authors
The authors gave an insight on the epidemiology, economic impacts, vaccine challenges, and future actions of PRRSV in Japan. This review is of certain significance for the prevention and control of PRRS in Japan, but the following revisions are still needed:
- The economic loss data is primarily based on a 2010 study and does not account for recent changes in farming practices or recurring epidemic impacts. We recommend supplementing with more recent data or providing justification for continuing to use this dataset. Given the higher pathogenicity of emerging strains (e.g., NADC34-like), it would be valuable to discuss whether their potential economic impacts have been assessed in relevant studies.
- Regarding the practical effectiveness of vaccines, while the limitations of MLV vaccines are mentioned, there lacks comparative analysis of different vaccines' actual protective efficacy in Japanese farms. We suggest citing local Japanese studies evaluating vaccine efficacy, particularly comparing key indicators such as neutralizing antibody titers and viral load suppression rates between vaccines like Ingelvac® and Fostera® against both endemic strains (e.g., Cluster IV) and emerging strains (e.g., NADC34-like).
- In the vaccine challenges section, while the shortcomings of MLV vaccines are discussed, we recommend further exploration of currently used vaccine types and coverage rates in Japan, including any progress in developing new vaccines, especially those targeting NADC34-like strains.
- For the regional control program section mentioning cases from Tahara City and Kagoshima Prefecture, we suggest supplementing with specific data such as post-implementation infection rate changes and economic benefits to strengthen the argument's persuasiveness.
- Some key content (e.g., global spread of NADC34-like strains, novel vaccine technologies) would benefit from additional references to important studies published within the last five years. For instance, including data on the global dissemination of NADC34-like strains and their potential impact on Japan, with citations from authoritative international journals to enhance the global perspective.
Author Response
Response to Reviewer 1 Comments
We extend our sincerest gratitude to reviewer #1 for the invaluable and thoughtful
evaluation of our manuscript (vetsci-366642) entitled "Epidemiological Review of Porcine
Reproductive and Respiratory Syndrome Virus (PRRSV) in Japan: From Discovery and
Spread to Economic Losses and Future Prospects ". We are deeply grateful for the recognition
of the potential significance of our review article in understanding the transmission of PRRSV,
which accompanies viral genomic RNA changes, and we genuinely appreciate the
constructive comments provided, which have been instrumental in refining our work. We
have diligently and carefully addressed each reviewer's concerns and suggestions, making
appropriate revisions to the manuscript with the aim of significantly enhancing its scientific
clarity and precision. Please find the point-by-point responses below, which explain the
changes made and clarifications offered. As instructed, we have endeavored to indicate the
corrections made in the manuscript text in red, to facilitate your review of the revisions.
Point-by-point response to Comments and Suggestions
Comment 1: The economic loss data is primarily based on a 2010 study and does not account for recent
changes in farming practices or recurring epidemic impacts. We recommend supplementing with more
recent data or providing justification for continuing to use this dataset. Given the higher pathogenicity
of emerging strains (e.g., NADC34-like), it would be valuable to discuss whether their potential
economic impacts have been assessed in relevant studies.
Reply to comment 1:
We are very grateful for your critical feedback regarding the economic loss data. As reviewer
#1 insightfully pointed out, the economic losses of approximately JPY 28 billion (USD 300
million) are based on the estimate by Yamane et al. in 2009. We concur with your observation
about the date of this data. To our knowledge, this remains the most comprehensive
nationwide estimate publicly available for Japan. It has consequently been used as a key
reference in domestic PRRS research in Japan because no other or revised estimation is
available in Japan. However, acting upon your valuable suggestion, we revised the sentence
to more explicitly clarify that this estimation was made more than 10 years ago
and therefore may not fully reflect the recent situation. (Page 6, Section 4, Lines 235-238)
Comment 2: Regarding the practical effectiveness of vaccines, while the limitations of MLV vaccines are
mentioned, there lacks comparative analysis of different vaccines' actual protective efficacy in Japanese
farms. We suggest citing local Japanese studies evaluating vaccine efficacy, particularly comparing key
indicators such as neutralizing antibody titers and viral load suppression rates between vaccines like
Ingelvac® and Fostera® against both endemic strains (e.g., Cluster IV) and emerging strains (e.g.,
NADC34-like).
Reply to comment 2:
We sincerely appreciate your valuable suggestion regarding the comparative analysis of
vaccine efficacy in Japanese farms. This was an excellent point that prompted us to enrich this
section. According to the reviewer’s suggestion, we have revised the manuscript to add further1
details and context describing the study of PRRSV vaccine usage in Japan. (Page 9, Section 7,
Lines 397-415)
Comment 3: In the vaccine challenges section, while the shortcomings of MLV vaccines are discussed,
we recommend further exploration of currently used vaccine types and coverage rates in Japan,
including any progress in developing new vaccines, especially those targeting NADC34-like strains.
Reply to comment 3:
Thank you for this insightful suggestion, which has helped us strengthen the manuscript
considerably. Regarding currently used PRRSV vaccines in Japan, we have listed the licensed
and utilized vaccines in Table 1. Following your guidance, we have added vaccine coverage
information obtained from a 2017 survey by the Japan SPF Swine Association. (Page 8, Section
7, Lines 361-369)
Additionally, we have comprehensively updated Section 8, subsection "3. Next-Generation
Vaccine Development and Evaluation," and added significant details, sentences, and
references that show progress in developing new PRRSV vaccines, especially those targeting
NADC34-like strains. (Pages 12-13, Lines 550-592)
Comment 4: For the regional control program section mentioning cases from Tahara City and
Kagoshima Prefecture, we suggest supplementing with specific data such as post-implementation
infection rate changes and economic benefits to strengthen the argument's persuasiveness.
Reply to comment 4:
Thank you for this very helpful suggestion. We wholeheartedly agree that specific
data indeed greatly strengthens the discussion on regional control programs. In direct
response to your feedback, we have substantially revised the descriptions of the field studies
carried out in Tahara City and Kagoshima Prefecture to incorporate much more
comprehensive details. (Page 11, Lines 489-508., Page 12, Lines 518-530)
Comment 5: Some key content (e.g., global spread of NADC34-like strains, novel vaccine technologies)
would benefit from additional references to important studies published within the last five years. For
instance, including data on the global dissemination of NADC34-like strains and their potential impact
on Japan, with citations from authoritative international journals to enhance the global perspective.
Reply to comment 5:
We are very grateful for your emphasis on the need to incorporate recent literature to
enhance the global perspective; this has undoubtedly improved the manuscript's relevance
and impact. According to the reviewer’s suggestion, we have thoroughly revised the
manuscript. Regarding the worldwide spread of NADC34-like strains, Section 6, "Incursion of
Foreign Strains and New Risks" has been significantly updated. In this revision, we included
more detailed information on their prevalence and impact in key regions such as China, U.S.,
South Korea, and Peru and cited recent studies mainly published within the last five years.
(Page 7, Lines 303-320)
Additionally, Section 8, subsection "3. Next-Generation Vaccine Development and
Evaluation" has been significantly updated to include novel vaccine technologies. The
discussion has now been revised to cover novel platforms, including mRNA vaccines, self-
amplifying RNA (saRNA) vaccines, refined subunit vaccines, strategies for broadly
neutralizing antibodies, chimeric viruses, and PROTAC technology. (Pages 12-13, Lines 550-
592)

Reviewer 2 Report
Comments and Suggestions for Authors
The authors gave a comprehensive review of the history of porcine reproductive and respiratory syndrome (PRRS) virus and the disease caused by it in Japan. PRRS is a very important disease all over the world, it has a great economic impact. It is an absolutely current issue, and the paper will be read with great interest. The paper is well written; it follows the general form of the review articles. The authors clearly indicate that they focus of PRRS in Japan but some internation outlook could be beneficial.
The authors should mention the immunosuppressive nature of the PRRS virus, how it can predispose to bacterial diseases.
What was the impact of the import of pigs on PRRS and the evolution of the virus?
What kind of factors helped the spread of the virus in Japan?
Author Response
Response to Reviewer 2 Comments
We extend our sincerest gratitude to reviewer #2 for the invaluable and thoughtful
evaluation of our manuscript (vetsci-366642) entitled "Epidemiological Review of Porcine
Reproductive and Respiratory Syndrome Virus (PRRSV) in Japan: From Discovery and
Spread to Economic Losses and Future Prospects ". We are deeply grateful for the
recognition of the potential significance of our review article in understanding the
transmission of PRRSV, which accompanies viral genomic RNA changes, and
we genuinely appreciate the constructive comments provided, which have been
instrumental in refining our work. We have diligently and carefully addressed each
reviewer's concerns and suggestions, making appropriate revisions to the manuscript with
the aim of significantly enhancing its scientific clarity and precision. Please find the point-
by-point responses below, which explain the changes made and clarifications offered. As
instructed, we have endeavored to indicate the corrections made in the manuscript text in
red, to facilitate your review of the revisions.
Point-by-point response to Comments and Suggestions
Comment 1: The authors should mention the immunosuppressive nature of the PRRS virus, how it
can predispose to bacterial diseases.
Reply to comment 1:
Thank you for this excellent suggestion. We fully agree that highlighting the
immunosuppressive nature of PRRSV and its association with secondary bacterial diseases
is crucial for a complete understanding of the virus's impact. According to the reviewer’s
suggestion, we have revised the manuscript to more thoroughly explain these issues. The
revised text now clearly shows that PRRSV-induced immunosuppression, particularly
through the impairment of porcine alveolar macrophage (PAM) function, renders pigs
highly susceptible to secondary bacterial and mycoplasmosis infections (e.g., Streptococcus
suis, Haemophilus parasuis, Mycoplasma hyopneumoniae) that leads to the exacerbation of
clinical disease as part of the Porcine Respiratory Disease Complex (PRDC).
(Page 2, Section 1, Line 76-83)
Comments 2: What was the impact of the import of pigs on PRRS and the evolution of the virus?
Reply to comment 2:
Thank you for this very pertinent comment. We believe our manuscript effectively
addresses the significant impact of pig importation on the introduction and evolution of
PRRSV in Japan at several key points. In Section 2, "Discovery of PRRSV and Characteristics
of Early Genotypes in Japan (1993–2007)", we described how the initial PRRSV strains were
predominantly of the North American type (Type 2), likely linked to imported breeding
stock. We also discuss the early presence and origins of different genetic clusters (e.g.,
Cluster III with East Asian connections, Cluster I with U.S. links, and the later emergence of
Cluster II associated with U.S./Asian strains and vaccine strains). Furthermore, subsequent
sections describe the incursion of European Type 1 virus (Section 3), virulent North
3American Lineage 1F strains, and the recent detection of NADC34-like strains (Section 6), all
of which underscore the continuous role of foreign introductions in shaping Japan's PRRSV
epidemiology. Throughout the manuscript, we have endeavored to illustrate how the
international movement of pigs (or related factors) has been a primary driver for the
introduction of new genetic diversity and the evolution of PRRSV in Japan. We believe these
discussions are indeed very much worth noting in the journal.
Comments 3: What kind of factors helped the spread of the virus in Japan?
Reply to comment 3:
Thank you for prompting us to further elaborate on the factors facilitating PRRSV spread in
Japan; this is a critical aspect of the epidemiological picture. We have already addressed
this in part in Section 8, "Future Actions and Prospects." Building on the existing statement
regarding the insufficiency of individual farm efforts in densely populated pig production
areas, this discussion explicitly addresses how Japan's geographical characteristics and the
concentration of pig production in certain regions contribute to this challenge.
However, inspired by your suggestion to provide even greater clarity, we added sentences
discussing that such high-density rearing environments can increase the risk of viral
transmission, provide more opportunities for viral persistence and evolution, and
potentially facilitate complex interactions between vaccine and field strains (e.g.,
recombination, spread of vaccine-derived viruses), thereby contributing to the genetic
diversification and spread of PRRSV. (Page 11, Lines 473-483)

Round 2
Reviewer 1 Report
Comments and Suggestions for Authors
The authors well responsed my conserns and suggestions and recommended for acceptance.